Culture of equine fibroblast-like synoviocytes on synthetic tissue scaffolds towards meniscal tissue engineering: a preliminary cell-seeding study

Warnock Jennifer J. 1 3 jennifer.warnock@oregonstate.edu
Fox Derek B. 1
Stoker Aaron M. 1
Beatty Mark 2
Cockrell Mary 1
Janicek John C. 1 4
Cook James L. 1
1 Comparative Orthopaedic Laboratory, University of Missouri , Columbia, MO , USA
2 VA Nebraska-Western Iowa Health Care System and University of Nebraska Medical Center College of Dentistry , Lincoln, NE , USA
Chin Wei-Chun
3 Current affiliation: College of Veterinary Medicine, Oregon State University, Corvallis, OR, USA

4 Current affiliation: Brazos Valley Equine Hospital, Navasota, TX, USA

Electronic publication date: 2014 Apr 17
Publication date: 2014
Volume: 2
Electronic Location ID: e353
Received 2014 Feb 2; Accepted 2014 Mar 28
Copyright: © 2014 Warnock et al.
Copyright year: 2014
Copyright holder: Warnock et al.
License: This is an open access article distributed under the terms of the Creative Commons Attribution License, which permits unrestricted use, distribution, reproduction and adaptation in any medium and for any purpose provided that it is properly attributed. For attribution, the original author(s), title, publication source (PeerJ) and either DOI or URL of the article must be cited.
License URL: https://creativecommons.org/licenses/by/4.0/

Keywords: Fibroblast-like synoviocytes, Cell scaffolds, Equine, Tissue engineering, Meniscus, Stifle, Bioreactors

Funding: Comparative Orthopaedic Laboratory, University of Missouri, Columbia, Missouri, USA This study was funded by the Comparative Orthopaedic Laboratory, University of Missouri, Columbia, Missouri, USA. The funders had no role in study design, data collection and analysis, decision to publish, or preparation of the manuscript.

==============================
Introduction. Tissue engineering is a new methodology for addressing meniscal injury or loss. Synovium may be an ideal source of cells for in vitro meniscal fibrocartilage formation, however, favorable in vitro culture conditions for synovium must be established in order to achieve this goal. The objective of this study was to determine cellularity, cell distribution, and extracellular matrix (ECM) formation of equine fibroblast-like synoviocytes (FLS) cultured on synthetic scaffolds, for potential application in synovium-based meniscal tissue engineering. Scaffolds included open-cell poly-L-lactic acid (OPLA) sponges and polyglycolic acid (PGA) scaffolds cultured in static and dynamic culture conditions, and PGA scaffolds coated in poly-L-lactic (PLLA) in dynamic culture conditions.

Materials and Methods. Equine FLS were seeded on OPLA and PGA scaffolds, and cultured in a static environment or in a rotating bioreactor for 12 days. Equine FLS were also seeded on PGA scaffolds coated in 2% or 4% PLLA and cultured in a rotating bioreactor for 14 and 21 days. Three scaffolds from each group were fixed, sectioned and stained with Masson’s Trichrome, Safranin-O, and Hematoxylin and Eosin, and cell numbers and distribution were analyzed using computer image analysis. Three PGA and OPLA scaffolds from each culture condition were also analyzed for extracellular matrix (ECM) production via dimethylmethylene blue (sulfated glycosaminoglycan) assay and hydroxyproline (collagen) assay. PLLA coated PGA scaffolds were analyzed using double stranded DNA quantification as areflection of cellularity and confocal laser microscopy in a fluorescent cell viability assay.

Results. The highest cellularity occurred in PGA constructs cultured in a rotating bioreactor, which also had a mean sulfated glycosaminoglycan content of 22.3 µg per scaffold. PGA constructs cultured in static conditions had the lowest cellularity. Cells had difficulty adhering to OPLA and the PLLA coating of PGA scaffolds; cellularity was inversely proportional to the concentration of PLLA used. PLLA coating did not prevent dissolution of the PGA scaffolds. All cell scaffold types and culture conditions produced non-uniform cellular distribution.

Discussion/Conclusion. FLS-seeding of PGA scaffolds cultured in a rotating bioreactor resulted in the most optimal cell and matrix characteristics seen in this study. Cells grew only in the pores of the OPLA sponge, and could not adhere to the PLLA coating of PGA scaffold, due to the hydrophobic property of PLA. While PGA culture in a bioreactor produced measureable GAG, no culture technique produced visible collagen. For this reason, and due to the dissolution of PGA scaffolds, the culture conditions and scaffolds described here are not recommended for inducing fibrochondrogenesis in equine FLS for meniscal tissue engineering.

Introduction

The knee menisci are semilunar-shaped fibrocartilages with extracellular matrix (ECM) composed primarily of types I and II collagen, glycosaminoglycans (GAGs), and water (Fithian, Kelly & Mow, 1990). It is now well established that intact menisci are crucial for the maintenance of normal joint function, however these critical structures are frequently injured in humans and animals. Meniscal tears are the most common knee injury in people, and arthroscopic meniscectomy represents the most common human orthopedic surgery performed annually (Burks, Metcalf & Metcalf, 1997). Meniscal injuries are also a significant cause of lameness and decreased performance in horses (Peroni & Stick, 2002; Walmsley, 1995; Walmsley, Phillips & Townsend, 2003); equines affected by naturally occurring meniscal tears may also be a viable model for the study of human meniscal injury.

As the axial, avascular portion of the meniscus has a limited ability to heal spontaneously, (Arnoczky & Warren, 1983; Kobayashi et al., 2004), the majority of meniscal injuries are treated with partial menisectomy. However, this also results in eventual articular cartilage damage of the tibia and femoral condyles, and progression of debilitating osteoarthritis (Arnoczky & Warren, 1983; Cox et al., 1975). Thus tissue engineering new meniscal fibrocartilage is being investigated as a treatment for avascular meniscal injuries.

Synovium may be an ideal cell source for meniscal tissue engineering. Synovium plays an important role in attempted vascular zone healing and regeneration (Cisa et al., 1995; Kobuna, Shirakura & Niijima, 1995; Ochi et al., 1996; Shirakura et al., 1997). Synovium has the ability to form fibrocartilaginous-like tissue in vivo in response to meniscectomy (Cox et al., 1975). In addition, synoviocytes have been reported to be an important element in cellular repopulation of meniscal allografts (Arnoczky & Warren, 1983; Rodeo et al., 2000). Synovial tissue progenitor cells, grossly indistinguishable in culture from type B or fibroblast-like synoviocytes (FLS), can undergo chondrogenesis in vitro (De Bari et al., 2001; Nishimura et al., 1999). Taken together, these data indicate that synovium may be able to serve as a source for functional fibrocartilage in engineering meniscal tissue, provided the chondrogenic potential of synoviocytes can be optimized.

Tissue engineering scaffolds must provide substrate and stability for cellular retention, intercellular communication, and cellular growth to allow seeded cells to proliferate extracellular matrix (ECM). As the scaffolds naturally degrade, the cellular ECM must be able to take on the biomechanical function and form previously designated by the scaffolds to maintain construct integrity. Thus a scaffold must be hydrophilic enough to allow cell adhesion but have a long enough half-life to not prematurely dissolve, which would prevent ECM proliferation and cell death. PGA (polyglycolic acid) and PLLA (poly-L-lactic acid) are biodegradable, biocompatible, polyesters, that are attractive for tissue engineering because they are readily available, can be easily processed into a variety of structures, and are approved by the Food and Drug Administration for a number of biomedical applications (Lavik et al., 2002). PGA has been successfully used as a scaffold for meniscal fibrochondrocytes in vivo (Kang et al., 2006) and cultured in vitro (Aufderheide & Athanasiou, 2005) to form meniscal-like tissue. PLLA has been successfully used for in vitro tissue engineering of leporine meniscal fibrochondrocytes (Esposito et al., 2013; Gunja & Athanasiou, 2010), chondrocytes (Sherwood et al., 2002), and human fibroblasts (Hee, Jonikas & Nicoll, 2006). PGA–PLLA combinations have also been successfully used for in vitro meniscal culture (Ionescu & Mauck, 2013). In addition, chondrocytes cultured on PGA-PLLA mixtures versus collagen sheets contain more collagen type II and have stronger mechanical properties (Beatty et al., 2002) than single polymer scaffolds. Further investigation of combination use of PLLA combined with PGA for in vitro synoviocyte culture is warranted.

Cartilage and fibrocartilage engineering with biodegradable scaffolds is most successful if uniform cell distribution is achieved (Davisson, Sah & Ratcliffe, 1999; Pazzano, Mercier & Moran, 2000; Smith, Dunlon & Gupta, 1995), which is optimized through the use of rotating bioreactors (Aufderheide & Athanasiou, 2005; Kim, Putnam & Kulik, 1998; Pazzano, Mercier & Moran, 2000). In addition, rotating bioreactors provide mechanical stimulation of cultured cells. This has a positive effect on cell differentiation, cell viability, extracellular matrix production, and compressive biomechanical properties, through mechanotransductive effects (Davisson, Sah & Ratcliffe, 1999; Imler, Doshi & Levenston, 2004; Pazzano, Mercier & Moran, 2000; Smith, Dunlon & Gupta, 1995). Thus scaffold culture in a rotating bioreactor may represent a useful technique for synoviocyte-based engineering of functional meniscal tissue.

Based on this prior research, we believe that both PGA and PLLA would be viable synthetic scaffolds for the in vitro culture of FLS for application in meniscal fibrocartilage tissue engineering. Thus, the first objective of this study was to (1) determine cell distribution and ECM formation of equine FLS seeded and cultured dynamically in a rotating bioreactor versus static seeding and culture, on two synthetic scaffold types, PGA and open-cell PLLA (OPLA). The second objective was to compare cell viability, distribution, and ECM formation of FLS cultured on 2% vs 4% PLLA coated PGA scaffolds, cultured for 14 or 21 days. Our hypothesis was that we would see no difference in equine FLS content, FLS distribution, and ECM formation between scaffold type, biomechanical culture environment, and culture duration.

Materials and Methods

Experiment 1

Tissue collection and monolayer cell culture

Six 8.0 mm × 8.0 mm biopsies of synovial intima and subintima were obtained from both stifles of an adult American Quarter Horse, euthanatized according the American Veterinary Medical Association’s guidelines for humane euthanasia, for reasons unrelated to the study. The horse was determined to be free of orthopedic disease based on pre-mortem physical examination and post mortem gross examination of the joint. Tissue was placed in Dulbeccos’ Modified Eagle’s Media (DMEM) with 10% fetal bovine serum, 0.008% Hepe’s buffer, 0.008% non-essential amino acids, 0.002% penicillin 100 I.U./mL streptomycin 100 ug/mL, amphoterocin B 25 ug/mL, 0.002% L-ascorbate, and 0.01% L-glutamine in preparation for monolayer culture.

Synovium was sectioned into 2.0 mm × 2.0 mm pieces using a #10 Bard Parker blade under sterile conditions. The tissue fragments were combined with sterile Type 1A clostridial collagenase solution (Type 1A Clostridial Collagenase; Sigma, St. Louis, MO) at a concentration of 7.5 mg/mL of RPMI 1640 solution. The mixture was agitated at 37 °C, 5% CO2, 95% humidity for six hours. Cells were recovered through centrifugation, the supernatant decanted and the cellular pellet re-suspended in 5 mL of supplemented DMEM. The cell solution was transferred to a 25 cm2 tissue culture flask containing 5 mL of supplemented DMEM. The flasks were incubated at 37 °C, 5% CO2, 95% humidity, with sterile medium change performed every 3 days. Synovial cells were monitored for growth using an inverted microscope until observance of 95% cellular confluence per tissue culture flask. At second passage cells were transferred to 75 cm2 tissue culture flasks containing 11 mL of media. At 95% confluence the cells were subcultured until the 4th cell passage had been reached. At 4th passage cells were removed from flasks, counted using the Trypan Blue exclusion assay (Strober, 2001), and transferred to scaffold culture as described below.

Scaffolds

A non-woven polyglycolic acid (PGA; Tissue Scaffold, Synthecon, Houston, TX) felt, 3 mm thick, with 10 µm diameter fibers was utilized for this study. The open-cell poly-lactic acid (OPLA sponge, BD Biosciences, Bedford, MA) utilized were 5.0 mm × 3.0 mm, non compressible, cylindrical sponges. The average OPLA sponge pore size was 100–200 µm with a hydration capacity of 30 µl/scaffold. PGA and OPLA scaffolds were sterilized in ethylene oxide. Following sterilization, the PGA felt was cut using a sterile Baker’s biopsy punch to create 5.0 mm diameter discs prior to cell culture.

Dynamic culture

Twelve PGA scaffolds (PGA-D group) and 12 OPLA sponges (OPLA-D group) were placed in separate 110 mL vessel flasks of a rotating bioreactor system (Rotating Bioreactor System, Synthecon, Houston, TX (Fig. 1) containing 110 mL of supplemented DMEM. The scaffolds were presoaked for 24 h in the bioreactor at 37 °C, 5% CO2, 95% humidity, prior to cell introduction. Fourth passage FLS were removed from the tissue culture flasks enzymatically (Accutase Innovative Cell Technologies, San Diego, CA) and counted. Cells were added to the 110 mL bioreactor flasks at a concentration of 1 million cells/scaffold via a 60 cc syringe, slowly injected over several minutes. For the duration of the study the bioreactor vessels were rotated at 51.1 rpm to allow the scaffolds to free-float and rotate within the culture medium, without contacting the inner bioreactor surfaces. Cultures were maintained at 37 °C, 5% CO2, 95% humidity. Fifty percent of the cell culture medium volume was changed using sterile technique every 3 days. Cell counts were performed on discarded media for the first two media changes.

Figure 1 Dynamic culture: rotating bioreactor apparatus.

Rotating wall bioreactor flask (110 mL) containing media and PGA scaffolds seeded with equine fibroblast-like synoviocytes (A). Flasks loaded on the rotating base apparatus; flasks rotate around their longitudinal axis (B).

Static culture

Twelve PGA scaffolds (PGA-S group) and 12 OPLA sponges (OPLA-S group) were placed individually in non-treated 24 well tissue culture plates, each well containing 2 mL of supplemented DMEM (Fig. 2). The scaffolds were presoaked for 24 h at 37 °C, 5% CO2, 95% humidity, prior to cell introduction. Then FLS were transferred from monolayer culture as described above, and slowly over 3 min, pipetted on top of the scaffolds in solution, at 1 million cells per scaffold in each well. The plates were maintained at 37 °C, 5% CO2, 95% humidity, with 50% cell culture medium changed every 3 days. Cell counts were performed on discarded media for the first 2 media changes.

Figure 2 Static culture of equine fibroblast-like synoviocytes on PGA scaffolds in a 24 well tissue culture plate, with each well containing 2 mL of supplemented DMEM.

Histologic analysis

All scaffolds were harvested on the 12th day of culture. Six scaffolds from each group (PGA-S, PGA-D, OPLA-S, OPLA-D) were fixed in 10% buffered formalin, embedded in paraffin, sectioned, and stained with Masson’s Trichrome, Safranin–O, and Hematoxylin and Eosin. Histologic specimens were examined at 10× magnification (Zeiss Microscope; Carl Zeiss, Thornwood, NY). Images of each section, (three from the scaffold periphery and three from the scaffold center) at 2 o’clock, 6 o’clock and 10 o’clock positions (Fig. 3) were digitally captured by a digital camera (Olympus DP-70 Olympus, Melville, NY) and saved as tagged-image file format images. Digital image analysis was performed as previously validated (Amin et al., 2000; Benzinou, Hojeij & Roudot, 2005; Girman et al., 2003; Goedkoop, Rie & Teunissen, 2005) whereby cellular density was assessed using a thresholding algorithm (Loukas et al., 2003) using computer image analysis (Fovea 3.0, Reindeer Graphics, Asheville, NC). This algorithm allows quantification of cellular nuclei based on their histogram values. All cell counts were additionally validated by hand counts. Safranin-O staining, indicating presence of GAG, and Masson’s Trichrome staining, indicating presence of collagen, were subjectively evaluated and recorded.

Figure 3 Histologic cell counting method.

Method for viewing all scaffolds to standardize cell counts and determine regional cell count differences between the scaffold center and periphery. Cells were counted at the periphery and central regions (dark dotted circles) of each scaffold (cross- hatched circle) using digital image analysis; peripheral cell counts (light dotted circles) were obtained at the 2 o’clock, 6 o’clock and 19 o’clock positions. Circles represent a low power (10× objective) field of view.

Biochemical ECM analysis

Three cultured scaffolds from each group were analyzed for glycosaminoglycan (GAG) and collagen production. Wet weight of each scaffold was obtained. GAG content of the scaffold was performed using the Dimethyl-methylene Blue Sulfated Glycosaminoglycan assay (Farndale, Buttle & Barrett, 1986). Collagen content of the cultured scaffolds was assessed using the hydroxyproline assay, as described by Reddy et al. (Reddy & Enwemeka, 1996).

Statistical methods

Data were tested for normality using a Shapiro–Wilk test. Data were then analyzed using a one way analysis of variance followed by a Tukey’s test, to compare the effect of scaffold type and seeding technique on cell counts and ECM quantity. To determine significance between periphery and central cell counts within each scaffold, a paired, 2-tailed student’s t-test was performed. For all tests significance was set at P < 0.05. All statistical analyses were performed using a statistical software program (GraphPad Prism Version 6, San Diego, CA).

Experiment 2

Scaffolds

PLLA was dissolved in methylene chloride as a 2% or 4% solution. The 2% and 4% PLLA solution each was applied to a 3.0 mm thick sheet of the same, above- described, non-woven PGA felt, using an eye-dropper. Following PLLA treatment, the treated felt was placed in a vacuum dessicator overnight and then sterilized in ethylene oxide. Following sterilization, the 2% and 4% PLLA modified PGA felts were cut into fourteen 5 mm × 7 mm × 3 mm square scaffolds using sterile scissors and a #10 bard parker blade (Fig. 4).

Figure 4 PLLA coated scaffolds.

Scanning electron microscopy of a 2% PLLA coated PGA scaffold (A) and a 4% PLLA coated scaffold (B) prior to cell seeding; bar = 100 μm.

Tissue collection and monolayer cell culture

Synovial intima/subintima was harvested from the stifles of two mixed breed, adult horses euthanatized according the American Veterinary Medical Association’s guidelines for humane euthanasia, for reasons unrelated to the study. These horses were also determined to be free of orthopedic disease based on pre-mortem physical examination and post mortem gross examination of the joint. The tissue was transported, minced and digested as described above. Cells were recovered through centrifugation, the supernatant decanted and the cellular pellet re-suspended in 5 mL of supplemented DMEM. The cell solution was transferred to a 25 mL tissue culture flask containing 5 mL of supplemented DMEM. Cells were grown in monolayer culture, under the conditions described above, until the 4th cell passage had been reached.

Dynamic culture

Fourteen 2% PLLA coated PGA scaffolds and fourteen 4% PLLA coated PGA scaffolds were placed in separate 110 mL vessel flasks of the rotating bioreactor system containing 110 mL of supplemented DMEM. The scaffolds were presoaked for 24 h in the bioreactor at 37 °C, 5% CO2, 95% humidity, prior to cell introduction. After this time it was noted that the scaffolds were floating at the apex of the flasks. Using sterile surgical technique, scaffolds were sterily removed from the flasks, pierced centrally, and strung on loops of 3-0 nylon surgical suture with knots placed adjacent to the scaffolds to prevent bunching on the line. Seven scaffolds were placed per suture. The strings of scaffolds were then placed back in to the bioreactors and presoaked for another 12 h, at which time complete hydration and submersion were achieved (Fig. 5).

Figure 5 Positioning of PLLA coated scaffolds in the rotating bioreactor.

Rotating wall bioreactor flask containing 2% PLLA coated PGA scaffolds, strung on suture to ensure equal submersion and positioning in the rotating flask.

Scaffolds were then dynamically seeded. Synovial membrane cells were removed from the tissue culture flasks using as described above and counted using the Trypan Blue exclusion assay (Strober, 2001). Cells were added to the bioreactor flasks at a concentration of 1 million cells/scaffold.

For the duration of culture, the bioreactor was maintained at 37 °C, 5% CO2, 95% humidity at 51.1 rpm. Fifty percent of the cell culture medium volume was changed using sterile technique every 3 days. Seven scaffolds were harvested on day 10 of culture, and 7 scaffolds were harvested on day 21 of culture.

Determination of cell viability

Cell viability was determined with the use of ethidium homodimer-1 (4 ul/ml PBS) and Calcein AM (Acetoxymethylester) (0.4 ul/ml PBS) fluorescent stains (Invitrogen, Carlsbad, CA) and the use of Confocal Laser Microscopy. The Confocal Laser Microscope consists of the BioRad Radiance 2000 confocal system coupled to an inverted microscope (Olympus IX70 Olympus, Melville, NY) equipped with Krypton–Argon and red diode laser. Approximately 1.0 mm sections were made from the halved scaffold using a rotary paper cutter. A section from each scaffold’s cut center and a section from each scaffold’s periphery was examined. Sections were incubated with the staining agents for 30 min at room temperature, placed on a glass microscope slide, moistened with several drops of PBS, and stained using the fluorescent double labeling technique. The sections were examined under 10 × magnification. Images were taken of each specimen as described above, (three from the section periphery and three from the section center) at the 2 o’clock, 6 o’clock and 10 o’clock positions. Images were digitally captured as described above. Live and dead cell counts were determined by hand counts.

DNA quantification

One half of each construct was lyophilized and a dry weight obtained. Samples were incubated in 1.0 ml Papain Solution (2 mM Dithiothreitol and 300 ug/ml Papain) at 60 °C in a water bath for 12 h. A double stranded DNA quantification assay (Quant-iT PicoGreen™ Invitrogen, Carlsbad, CA) was performed. Double stranded DNA extracted from bovine thymus was mixed with TE buffer (Invitrogen, Carlsbad, CA) to create standard DNA concentrations of 1,000, 100, 10, and 1 ng/mL. The standards and 100 uL of each papain digested sample (used in the above GAG and hydroxyproline assays) were added to a black 96 well plate. 100 uL of 2 ug/mL of Pico Green reagent was added to each well and the plate was incubated for 5 min. Sample fluorescence was read at 485 nm excitation/528 nm emission by a spectrophotometric plate reader (Synergy HT–KC-4; BioTek, Winooski, VT). Absorbances were converted to ng/mL concentrations and total double stranded DNA yield in ng using FT4 software (BioTek, Winooski, VT).

Statistical methods

Data were tested for normality using a Shapiro–Wilk test. Scaffold weights were compared using a 2-tailed paired t-test. Scaffold dsDNA content was analyzed using a repeated- measures analysis of variance with a Geisser-Greenhouse correction. Significance was set at p < 0.05. All statistical analyses were performed using a statistical software program, (GraphPad Prism Version 6, San Diego, CA).

Results

Experiment 1

As determined by the Trypan Blue exclusion assay, viability of cells at the time of transfer from monolayer culture to static or dynamic seeding was 98.6%. No live cells were detected in any of the media changes for either static or dynamically cultured scaffolds, indicating that viable cells rapidly adhered to the scaffolds.

At the time of harvest upon gross examination, the fibers of the PGA scaffolds and the sponge surface of the OPLA scaffolds were still visible. PGA scaffolds subjectively appeared more translucent.

Despite equal cell seeding concentrations, the effect of dynamic bioreactor culture on cell content of PGA scaffolds (PGA-D versus PGA-S) was to increase scaffold cellularity (P < 0.001). This was also found in OPLA-D versus OPLA-S scaffolds (P = 0.028). The effect of scaffold type also significantly increased scaffold cellularity of PGA-D versus OPLA-D (P = 0.017), while OPLA-S had great cellularity than PGA-S (P = 0.0217; Table 1).

Table 1 The effect of seeding and cell culture biomechanical environment and the effect of scaffold type on scaffold cellularity.

	Cell count (Mean number of cells per 10× objective field ±SD)	Effect of biomechanical environment (dynamic vs static culture)	
Scaffold type	Biomechanical environment		
	Dynamic culture	Static culture		
PGA	1128 ± 575 cells	54 ± 34 cells	P < 0.001	
OPLA	375 ± 118 cells	301 ± 65 cells	P = 0.028	
Effect of scaffold type (PGA vs OPLA)	P = 0.017	P = 0.0217		

All groups, with the exception of OPLA-S, showed increased cellular distribution to the periphery of the scaffolds (Table 2). Due to the shape of the OPLA-S on histological sectioning, there was overlap of central and peripheral fields of view, precluding accurately localized cell counts; peripheral cell count was 307 ± 52 and central cell count was 287 ± 80 (P < 0.464). Cells grew in whorls, strands, and sheets on the PGA scaffolds, while cells grew in clumps on the surface pores of the OPLA sponges (Fig. 6).

Figure 6 Histologic cell distribution on PGA and OPLA scaffolds.

Micrographs of scaffolds seeded with equine fibroblast-like synoviocytes; Hematoxylin and Eosin staining, 10× objective magnification; bar = 100 μm. (A) PGA scaffold cultured in a static environment; (B) PGA scaffold cultured in a dynamic environment (rotating bioreactor); (C) OPLA scaffold cultured in a dynamic environment (rotating bioreactor); (D) OPLA scaffold cultured in a static environment. Note the intact PGA fibers (open arrow) and the cells located in clumps in the pores of the OPLA scaffold (closed arrows).

Table 2 Peripheral and central cell count (Mean number of cells per 10× objective field ±SD).

Scaffold	Peripheral cell count	Central cell count	P-value	
PGA-D	1433 ± 487	724 ± 314	P < 0.001	
PGA-S	80 ± 28	28 ± 11	P < 0.001	
OPLA-D	476 ± 90	295 ± 55	P < 0.001	

Staining for collagen and glycosaminoglycan using Masson’s Trichrome and Safranin-O, respectively, was negative for extracellular matrix production in all sections of all scaffold types and culture conditions evaluated.

In the PGA-D group, the dimethylmethylene blue assay detected a mean of 22.29 µg of GAG per scaffold, (range 19.34–28.13 µg), with a mean % GAG scaffold content of 0.0345% (µg GAG per µg scaffold wet weight). No GAG was detected in OPLA constructs or PGA-S constructs. The hydroxyproline assay did not detect collagen production in any group.

Experiment 2

Post PLLA modification, mean scaffold dry weights before soaking and seeding were 1.01 mg for 2% PLLA coating and 1.52 mg for 4% PLLA coating (P < 0.001). Scaffold dry weights decreased over time. Mean lyophilized weight on day 10 for 2% PLLA coating was 0.533 mg, which decreased to 0.257 mg on day 21 (P = 0.02). Mean lyophilized weight on day 10 for 4% PLLA coating was 0.481 mg, which decreased to 0.381 mg on day 21 (P = 0.043).

Scaffold cellularity as measured by dsDNA content increased over time: for the 2% group, day 10 cellularity was 102.6 ng dsDNA/mg dry weight, and on day 21 it was 281.79 (P = 0.021). On day 10 for the 4% group, dsDNA content was 111.01 ng dsDNA/mg dry weight and on day 21 it was 140.2 ng dsDNA/mg dry weight (P = 0.032; Fig. 7).

Figure 7 Double stranded DNA content of PLLA coated scaffolds.

Mean ± Standard Error of the Mean (SEM) of dsDNA content of PGA scaffolds coated in 2% PLLA and 4% PLLA, seeded dynamically and cultured in a rotating bioreactor for 14 days and 21 days. A bar and (*) indicates a significant difference between two treatment groups (P < 0.05).

PLLA coating also affected scaffold dsDNA content. Scaffolds with the 2% PLLA coating had greater dsDNA content than the 4% PLLA coating on day 21 (P = 0.003), but not on day 10 (P = 0.602; Fig. 7).

As visible under confocal microscopy, cells only adhered to the surface of exposed PGA fibers and had poor to no penetration to the scaffold centers in all PLLA coated scaffolds. Viable cell numbers were estimated only because of the marked cellular clumping; all scaffolds showed mixtures of viable and non-viable cells localized in clumps on the scaffold outer margins (Figs. 8–11). Histologic examination of H + E stained constructs revealed minimal cellular adhesion to the PLLA, in all groups at all times, with cells growing primarily on the exposed PGA scaffold, in tightly packed clumps, or adhering to exposed fibers of PGA. No extracellular matrix was observed in any scaffolds on histologic analysis, which also reflected the uneven cellularity (Fig. 12).

Figure 8 Cell viability: 2% PLLA scaffolds.

Photomicrographs of 2% PLLA coated PGA constructs harvested on day 10, under standard light (column A) and under laser confocal microscopy (column B), using the calcein AM-ethidium homodimer live-dead assay. Images represent scaffold transverse cross sections (row T) and scaffold surface coronal sections (row C). Green stained cells are alive, red stained cells are dead. 10× objective magnification; bar = 100 μm.

Figure 9 Cell viability: 2% PLLA scaffolds.

Photomicrographs of 2% PLLA coated PGA constructs harvested on day 21, under standard light (column A) and under laser confocal microscopy (column B), using the calcein AM-ethidium homodimer live-dead assay. Images represent scaffold transverse cross sections (row T) and scaffold surface coronal sections (row C). Green stained cells are alive, red stained cells are dead. Note the spurious red staining of scaffold PGA fibers. 10× objective magnification; bar = 100 μm.

Figure 10 Cell viability assay: 4% PLLA scaffolds.

Photomicrographs of 4% PLLA coated PGA constructs harvested on day 10, under standard light (column A) and under laser confocal microscopy (column B), using the calcein AM-ethidium homodimer live-dead assay. Images represent scaffold transverse cross sections (row T) and scaffold surface coronal sections (row C). Green stained cells are alive, red stained cells are dead. Note the spurious red staining of PGA fibers. 10× objective magnification; bar = 100 μm.

Figure 11 Cell viability: 4% PLLA scaffolds.

Photomicrographs of 4% PLLA coated PGA constructs harvested on day 21, under standard light (column A) and under laser confocal microscopy (column B), using the calcein AM-ethidium homodimer live-dead assay. Images represent scaffold transverse cross sections (row T) and scaffold surface coronal sections (row C). Green stained cells are alive, red stained cells are dead. Note the spurious red staining of PGA fibers. 10× objective magnification; bar = 100 μm.

Figure 12 Distribution of cells on 2% and 4% PLLA coated PGA scaffolds.

Photomicrographs of 2% PLLA coated PGA scaffolds harvested on day 10 (row 1) and day 21 (row 2), and 4% PLLA coated PGA scaffolds harvested on day 10 (row 3) and day 21 (row 4), H+E staining. Column A represents images of the center of the construct and column B represents images taken of the scaffold periphery. Note that the cells have grown in dense clusters; 10× objective magnification; bar = 100 μm.

Discussion

The current study analyzed the effect of scaffold type, biomechanical stimuli, and culture duration on FLS seeding and production of specific meniscal ECM constituents. We found that FLS-seeded PGA constructs cultured in a rotating bioreactor had the highest cellularity, with a mean sulfated glycosaminoglycan content of 22.3 µg per scaffold. PGA constructs cultured in static conditions had the lowest cellularity. For PLLA coated PGA, increasing concentration of PLLA decreased scaffold cellularity, while increased culture time increased scaffold cellularity, as determined by the dsDNA assay. A non-uniform cellular distribution was observed for all scaffold types and culture conditions.

Bioreactor culture provides a number of benefits over static culture which would account for the higher cellularity of PGA-D and OPLA-D versus PGA-S and OPLA-S scaffolds. The rotating wall bioreactor used in this study provided a dynamic, laminar fluid shear, which perfuses scaffold cultured cells (Bilodeau & Mantovani, 2006), and thereby encourages cell survival and proliferation by providing efficient transport of nutrients, gases, catabolites, and metabolites and maintaining physiologic media pH (Gooch et al., 2001; Vunjak-Novakovic et al., 1998). Mixing of culture media also promotes cell seeding by creating matched relative velocities of cells and scaffolds, particularly on non-woven PGA scaffolds (Vunjak-Novakovic et al., 1998). In addition, the rotating wall bioreactor limits cellular stress by reducing strong shear forces and cellular impact on the walls of the bioreactor (Bilodeau & Mantovani, 2006). However, in our study, scaffold characteristics such as scaffold density and hydrophilicity may have negated the advantages of bioreactor culture, as seen with OPLA or PLLA coated scaffolds, which had fewer cells and markedly uneven cell distribution, respectively.

A higher cell count was found on PGA-D versus OPLA-D, indicating either better adherence or cell proliferation on PGA. Non-woven PGA scaffolds favor cellular capture and retention because of their polar surface properties and high surface area for cellular adhesion (Day, Boccaccini & Shurey, 2004; Moran, Pazzano & Bonassar, 2003). Cellularity of PGA-D was further increased by the open weave and low density (45–77 mg/cc) of PGA scaffolds supports cellular proliferation through superior flow-through of culture media and nutrient delivery (Vunjak-Novakovic et al., 1998). This is in contrast to the highly dense (871 mg/cc) OPLA sponges with non-communicating pores, which could inhibit nutrient and gas transfer to seeded cells (Pazzano et al., 2004; Pazzano, Mercier & Moran, 2000; Wu, Dunkelman & Peterson, 1999). For PLLA covered PGA scaffolds, cells were located primarily on exposed PGA fibers, and scaffold cellularity was inversely proportional to the concentration of PLLA. Although PLLA is widely used in tissue-engineering applications because of its slower degradation characteristics, strength, and mechanical properties, its hydrophobic, inert nature can affect cell–matrix interactions and decrease cellular adhesion (Moran, Pazzano & Bonassar, 2003). While the PLLA coating of PGA scaffolds was intended to protect from premature scaffold dissolution, we observed that with longer duration of culture, scaffolds appeared to be more fragile to disruption with forceps manipulation, particularly on the outer edges as well as around the centrally placed suture. In agreement with this observation, all scaffold dry weights dropped over time, indicating scaffold dissolution. Thus PLLA did not prevent PGA hydrolysis and decreased scaffold integrity. PLLA coating also provided a hydrophobic barrier to centralized cell seeding and ingrowth. Thus, for the future study of scaffold seeded equine FLS, use of PLLA type scaffolds is not recommended.

Cell distribution across all scaffolds was uneven, in contrast to previous reports on bioreactor chondrocyte culture (Mahmoudifar & Doran, 2005; Pazzano et al., 2004). Lower central cell density in our scaffolds may have indicated poor axial cell penetration and in-growth. Alternatively, higher peripheral cellularity could reflect increased peripheral cell division caused by increased exposure to media nutrients, gas exchange, and mechanotransductive effects (Mahmoudifar et al., 2002). Additionally the OPLA scaffolds had clumped cell distribution in the outermost pores. The OPLA sponge porosity may not allow uniform cell distribution; the 100–200 µm pores do not consistently communicate with each other. While OPLA-S did not have different peripheral and central cell counts, this was due to an artifact of the sponge shape and precluded distinction of peripheral cells from central cells. To increase central scaffold cell content, flow-through bioreactors (Bilodeau & Mantovani, 2006) may have greater cell seeding efficiencies than rotary bioreactors. Alternatively, cells may be seeded at the time of scaffold formation, such as during hydrogel synthesis, to insure central scaffold cellularity (Narita et al., 2009).

The culture conditions utilized in the present study resulted in minimal to no ECM formation, in contrast to other studies. The mean GAG content of the PGA-D scaffold of 0.0345% (wet weight basis) was lower than the 0.6–0.8% wet weight in the normal meniscus, and thus represents a sub-optimal response for engineering purposes (AufderHeide & Athanasiou, 2004). Synoviocytes typically produce collagen type I constitutively, (Garner, 2000; Levick, Price & Mason, 1996), however production and deposition of hydroxyproline was not detected in this study. The most likely reason for this failure of ECM formation was lack of culture with a specific fibrochondrogenic media. For example, culture with recombinant transforming growth factor-beta, insulin-like growth factor-1 and basic fibroblast growth factor have been shown to induce in vitro collagen formation in human synoviocytes (Pei, He & Vunjak-Novakovic, 2008; Pei et al., 2008). Treatment of equine FLS with recombinant chondrogenic growth factors, in addition to the scaffold and bioreactor culture conditions used in the present study, resulted in greater type II collagen and aggrecan gene expression (Fox et al., 2010). Reported scaffold seeding concentrations for cartilage tissue engineering include 30,000 fibroblasts/mL (Day, Boccaccini & Shurey, 2004); 600,000 chondrocytes/mL (Stading & Langer, 1999); 5 million chondrocytes/mL (Griffon, Sedighi & Sendemir-Urkmez, 2005); and 10 million chondrocytes/mL (Hu & Athanasiou, 2005). Our seeding density of 1 million equine FLS per scaffold may have been too low, as dense cell aggregates are required for meniscal developmental fibrochondrogenesis (Clark & Ogden, 1983) due to the embryonic community effect (Gurdon, Lemaire & Kato, 1993). In the present study the FLS were exposed to the mild shear forces and hydrostatic pressurization in a rotating bioreactor (Mauck et al., 2002) which may not have been the optimal type of forces required for synovial collagen I formation. A combination of in vitro tensile and compressive forces (AufderHeide & Athanasiou, 2004; Benjamin & Ralphs, 1998) may be required to support formation GAG (Valiyaveettil, Mort & McDevitt, 2005) and types I and II collagen (Kambic & McDevitt, 2005), the major ECM components of fibrocartilage. Cell culture on scaffolds may also result in cellular stress shielding, thereby resulting in suboptimal matrix formation (Huey & Athanasiou, 2011). Synovial macrophages may have contaminated our FLS cultures, thereby also decreasing ECM formation (Pei, He & Vunjak-Novakovic, 2008; Bilgen et al., 2009) and future studies should include negative isolation of macrophages. Additionally, co-culture with meniscal fibrochondrocytes as decribed by Tan and workers (Tan, Zhang & Pei, 2010) may have also helped fibrochondrogenic differentiation of equine FLS and will be the focus of future studies. Increased culture time may also be beneficial to ECM formation; other studies show time dependent ECM expression (Griffon, Sedighi & Sendemir-Urkmez, 2005; Mueller et al., 1999; Sha’ban et al., 2008). One study of synovial chondrogenesis on PGA scaffolds utilized a longer culture duration of 60 days, with successful ECM formation (Sakimura et al., 2006). Despite better cellularity, PGA scaffolds began losing integrity over the culture period, even when coated with PLLA. Unless rapid ECM formation can be achieved before dissolution occurs, PGA hydrolyzes too quickly (t1/2 = 16 days) for the purpose of long term meniscal fibrocartilage synthesis. Treatment with chondrogenic or fibrochondrogenic media may induce production of ECM, thus making the culture systems described here more feasible for meniscal tissue engineering.

Conclusion

In conclusion, we reject the null hypothesis; dynamic cell seeding and culture, as well as increased culture duration, increased scaffold cellularity. Scaffold type also affected cellularity; for bioreactor culture, PGA had higher cell counts versus OPLA, while OPLA had higher cell counts versus PGA in static culture. Cells could only grow unevenly in the pores of the OPLA sponge, and cells could not adhere to the PLLA coating of PGA scaffolds. Increasing the concentration of PLLA coating on a PGA scaffold decreased the cellularity of the scaffold, and did not prevent scaffold dissolution. While PGA culture in a bioreactor produced measureable GAG, no culture technique produced visible collagen. For this reason, and due to the dissolution of PGA scaffolds, the exact culture of conditions described here are not recommended for inducing equine fibrochondrogenesis towards meniscal tissue engineering. Further research is recommended to enhance extracellular matrix production through additional biomechanical and biological stimulation, including treatment with chondrogenic media, increased culture duration, and increased cell seeding concentrations.

Additional Information and Declarations

Competing Interests

Author Contributions

The authors of this manuscript have no financial, professional, personal, or other competing interests to declare, which would have otherwise caused bias regarding this work.

Jennifer J. Warnock performed the experiments, analyzed the data, wrote the paper, prepared figures and/or tables, reviewed drafts of the paper.

Derek B. Fox conceived and designed the experiments, reviewed drafts of the paper.

Aaron M. Stoker performed the experiments, analyzed the data, reviewed drafts of the paper.

Mark Beatty conceived and designed the experiments, performed the experiments, contributed reagents/materials/analysis tools, reviewed drafts of the paper.

Mary Cockrell performed the experiments, reviewed drafts of the paper.

John C. Janicek performed the experiments, contributed reagents/materials/analysis tools, reviewed drafts of the paper.

James L. Cook conceived and designed the experiments, contributed reagents/materials/analysis tools, reviewed drafts of the paper.

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
