# Peer review of "Culture of equine fibroblast-like synoviocytes on synthetic tissue scaffolds towards meniscal tissue engineering: a preliminary cell-seeding study"

_PeerJ, doi:10.7717/peerj.353_

## Round 0.1 · original submission · Major Revisions

Reviewer #1 suggests that the medium use is the reason for the reported observation that is different from authors' conclusion. Please address this difference.

Reviewer 1 ·

Basic reporting

see general comments

Experimental design

see general comments

Validity of the findings

see general comments

Additional comments

The paper under review aimed at evaluation of feasibility to seeding fibroblast-like synoviocytes on synthetic scaffolds, including PGA, PLLA coated PGA, and OPLA, in either rotating bioreactor or 24-well plate for meniscal tissue engineering. The authors found that while PGA culture in a bioreactor produced measureable GAG, no culture technique produced visible collagen. Thus the conclusion is the culture conditions and scaffolds described here are not recommended for inducing fibrochondrogenesis in equine FLS for meniscal tissue engineering. I believe the failure of this project was due to the lack of chemically defined chondrogenic medium during the incubation of cell-scaffold constructs. My other concerns include (1) too much literatures were cited in both Introduction and Discussion and made this paper look like a review paper, (2) it’s not clear how much cells were added into a bioreactor, and (3) whether the cell number per scaffold in bioreactor was comparable with that in 24-well plate.

Reviewer 2 ·

Basic reporting

A few editorial issues: In the materials and methods of the abstract there should be a space between PGA and and.
The word Results in the abstracted should be bolded.
In the text of Figure 10 in the 4th line in the paragraph, there should be a space between are and alive.

In the Experiment 2 section of the results: lines 288-291 "subjectively, with longer duration of culture............" I believe this belongs in the discussion. I think it should read that the weights were lower, and then this can be expanded on in the discussion.

Experimental design

Well thought out and well executed study. No complaints.

Validity of the findings

No comments.

Additional comments

No comments. Great study.

---

## Round 0.2 · Minor Revisions

Please address the comment from Reviewer #1 and revise the manuscript.

Reviewer 1 ·

Basic reporting

none

Experimental design

none

Validity of the findings

none

Additional comments

The revision made by the authors improves the quality of the paper. However, a few concerns still exist as below.

1. In the response from the authors, Line 359 now reads “….However, treatment of equine FLS with the same recombinant chondrogenic growth factors, in the same culture conditions as the present study, also failed to produce measureable ECM (Fox et al. 2009)”. This statement is not consistent with the conclusion from the original paper (Fox et al. 2010). Here is the text from the published abstract “…Cultured FLS were seeded onto synthetic scaffolds in a rotating bioreactor under the influence of three growth factor regimens: none, basic fibroblast growth factor (bFGF) alone, and bFGF plus transforming growth factor (TGF-beta(1)) and insulin-like growth factor (IGF-1). Constructs were analyzed for mRNA expression and production of fibrochondroid extracellular matrix constituents. Type II collagen and aggrecan mRNA were significantly higher in growth factor-treated groups (p<0.05)”. Obviously, treatment of FLS with chondrogenic growth factors benefited to produce measurable ECM. Another mistake is this paper (Fox et al.) published in 2010 rather than 2009.
2. Despite the fact that this is a failure study, the weakness in this study deserves a full discussion. For example, the contamination of macrophages (type A synovial cell) could be responsible for the loss of matrix (Bilgen et al., CD14-negative isolation enhances chondrogenesis in synovial fibroblasts. Tissue Eng Part A. 2009 Nov;15(11):3261-70) (Pei et al. Engineering of functional cartilage tissue using stem cells from synovial lining: a preliminary study. Clin Orthop Relat Res. 2008 Aug;466(8):1880-9). In addition, the co-culture with meniscus cells might help guide FLS toward meniscus-like cells (Tan et al., Meniscus reconstruction through coculturing meniscus cells with synovium-derived stem cells on small intestine submucosa--a pilot study to engineer meniscus tissue constructs. Tissue Eng Part A. 2010 Jan;16(1):67-79).

Minor:

1. In Page 9 on line 193, two “PPLA”s should be “PLLA”s.

Reviewer 2 ·

Basic reporting

No complaints

Experimental design

No complaints

Validity of the findings

No complaints

---

## Round 0.3 · accepted · Accept

You have addressed the concern of reviewers in your revision.